# Key Recovery for Content Protection Using Ternary PUFs Designed with Pre-Formed ReRAM

**Bertrand Francis Cambou *** and **Saloni Jain**

Department of Applied Physics and Material Science, College of Engineering Informatics and Applied Sciences, Northern Arizona University, Flagstaff, AZ 86011, USA; sj799@nau.edu
* Correspondence: Bertrand.cambou@nau.edu

**Featured Application: Protection, and secure delivery of digital files in a network of terminal devices connected to a service provider.**

**Abstract:** Physical unclonable functions, embedded in terminal devices, can be used as part of the recovery process of session keys that protect digital files. Such an approach is only valuable when the physical element offers sufficient tamper resistance. Otherwise, error correcting codes should be able to handle any variations arising from aging, and environmentally induced drifts of the terminal devices. The ternary cryptographic protocols presented in this paper, leverage the physical properties of resistive random-access memories operating at extremely low power in the pre-forming range to create an additional level of security, while masking the most unstable cells during key generation cycles. The objective is to reach bit error rates below the $10^{-3}$ range from elements subjected to drifts and environmental effects. We propose replacing the error correcting codes with light search engines, that use ciphertexts as helper data to reduce information leakage. The tamper-resistant schemes discussed in the paper include: (i) a cell-pairing differential method to hide the physical parameters; (ii) an attack detection system and a low power self-destruct mode; (iii) a multi-factor authentication, information control, and a one-time read-only function. In the experimental section, we describe how prototypes were fabricated to test and quantify the performance of the suggested methods, using static random access memory devices as the benchmark.

**Keywords:** key recovery; content protection; cybersecurity; unclonable devices; network of IoT; tamper resistance; error correction

---

## 1. Introduction

Physical elements, physical unclonable functions (PUFs), tags, and other hardware systems have been proposed to secure both terminal devices and their digital files because they can be tamper resistant. Considering the economic importance of such developments in the field of information technology, the systems are often protected by patents, as seen with these recent patents [1–4]. This list is for reference purposes only, and is not exhaustive, given the breath of the topic under consideration. In [1], the recovery of a root key from the measurement of a circuit function with a PUF is proposed. A checkpointing feature is used to periodically mark measurements of this function and track drift in the value of the root key over the life of a digital device. In [2], PUF-based devices are proposed to distribute encrypted messages to a control device connected through a wide-area communication network. In [3], systems and methods are proposed to improve a computer system's resistance to tampering. Most components of the system do not have the same level of protection against tampering as the PUF. The tamper protection provided by the PUF may be extended to other components of the system, thus creating a network of tamper-resistant components. In [3], the system includes a tamper detection circuit that receives signals from the component(s). The tamper detection circuit generates an output signal which indicates

whether any of the components have been tampered with. If the output signal indicates that one of the components has been tampered with, the PUF corrects and mitigates the loss of secret information. Finally, [4] describes a system for recording digitally signed files using an authorization token.

In this study, our interest is in terminal devices that are part of networks of the internet of things (IoTs), and which interact with a service provider. We propose end-to-end solutions that are based on the recovery of cryptographic keys from tamper-resistant PUFs, as a means of protecting the delivery and storage of digital files in a network of IoTs and terminal devices, driven by a service provider. This paper is organized as follows:

[Section 2] In this section we present a generic description of one-way unclonable functions and physical unclonable functions. We also present how the use of ternary states has the potential to reduce the bit error rates in the part per million range (ppm).

[Section 3] The architecture allowing the secure key recovery from the ternary PUFs, is shown in Section 3. The replacement of mainstream error correcting codes (ECC) by a search engine such as response-based cryptography (RBC) is suggested. We explain how the helper data needed by ECC is replaced by a message digest of the key that does not leak information.

[Section 4] The overall architecture allowing the encryption and decryption of the stored digital files with the session key, is shown in Section 4. We present variations that incorporate public key infrastructures (PKIs).

[Section 5] In this section we detail how the ternary PUFs can be implemented with resistive random-access memories (ReRAM) operating in the pre-forming range. We suggest methods to exploit their physical properties to enhance tamper resistance, to sense certain attacks, and to self-destruct the device, at low power, when needed.

[Section 6] In Section 6, the experimental work conducted to validate the concept is presented. A full prototype with custom ReRAM circuits allows the characterization and optimization of the solutions in terms of latencies and bit error rates. The cryptographic algorithms selected for this study are SHA-3, SHAKE, and elliptic curves, and the algorithms under consideration for standardization for the post-quantum cryptography are by NIST.

## 2. One Way Unclonable Functions with Ternary States

### 2.1. One Way Unclonable Functions

The one-way unclonable functions "**Ψ**" in this paper, are defined as functions generating a stream of bits "**K**" from a random number "**T**", and individual digital access instructions "***IDAccess***", as shown in Equation (1):

$$\mathbf{K} \leftarrow \mathbf{\Psi}\ (\mathbf{T}, \textit{\textbf{IDAccess}}) \tag{1}$$

The function **Ψ** is kept secret in such a way that the knowledge of **T** and ***IDAccess*** does not disclose **K**; therefore, it is assumed that (**T**, ***IDAccess***) can become public information.

2.1.1. Random Number (**T**)

- The random number **T** feeds an extended output function (XOF) pointing at a set of addresses **A** contributing to the generation of stream K. For example, the XOF can be a SHAKE from the message digest **MD** of the SHA-3 hashing function.
- **T** can be concatenated with password **PW**, as shown in Equation (2):

$$\mathbf{A} \leftarrow \mathrm{XOF} \leftarrow \mathbf{MD} \leftarrow \mathrm{Hash}\ (\mathbf{T} \oplus \mathbf{PW}) \tag{2}$$

- With the use of a password, or another multi-factor scheme, **T** can be freely disclosed through insecure communication channels.

2.1.2. Individual Digital Access Instructions (*IDAccess*):

- The individual digital access *IDAccess* is used to retrieve the set of instructions "**I**" needed to generate **K** from the set of addresses **A**. To enhance security, **I** can be XORed with the message digest "**MD**" of the XOF as shown in Equation (3):

$$I \leftarrow \text{Hash} (IDAccess \oplus \text{MD}) \tag{3}$$

- With this protection, *IDAccess* can be freely disclosed through insecure communication channels.
- The set of instructions **I** can incorporate is a ternary representation that reduces the bit error rates (BER) of the output stream **K**, and it can offer additional protection. When an opponent tests the one-way function without knowing the position of the ternary states, **Ψ** generates streams with high BER and potentially damages the structure permanently.

2.1.3. One-Way-Ness of the Function **Ψ**

- The knowledge of **K** does not disclose the input parameters (**T**, *IDAccess*).
- The knowledge of one input parameter alone, **T** or *IDAccess*, does not disclose **K.**

2.1.4. Collision Avoidance

- Any change in the input parameters is likely to generate different output.
- Two different outputs are most likely the result of different inputs unless the difference in the output is small enough.
- Repeating the function **Ψ** could result in small variations of the **K** stream; let us say that typically 90% of the stream will be the same.

2.1.5. Un-Clonability

- The function is unclonable and can have a physical execution, making it highly unlikely to be duplicated.
- During "enrollment", the image of the one-way function of the client device can be downloaded in a look-up table of the controlling device. This allows the controlling device to communicate safely with the client device as both parties can independently generate the same stream **K** from the shared input parameters **T** and *IDAccess*, and then they can use **K** as part of a cryptographic protocol.

*2.2. One Way Unclonable Functions with PUFs*

The one-way unclonable functions can be implemented with PUFs, exploiting nanocomponents that are unique and unclonable due to small variations occurring in their fabrication. PUFs are currently used for both authentication and key generation. PUFs are described by the one-way function of $f$ converting $n$-bit challenges $\mathbf{C} = \{c_1; \ldots ; c_i; \ldots ; c_n\}$ in $m$-bit responses $\mathbf{K} = \{k_1; \ldots ; k_j; \ldots ; k_m\}$; $c_i$ and $k_j \in \{0, 1\}$:

$$\mathbf{K} = f(\mathbf{C}) \leftarrow \mathbf{C} \tag{4}$$

$\mathbf{C} = (\mathbf{T}, IDAccess)$ of Section 2.1 is the stream of challenges, while **K** is the stream of responses. During key generation cycles, the responses of **K** should match the ones generated upfront during enrollment. The protocols based on PUFs are effective with error-correcting schemes that are able to handle moderate aging, and environmental effects [5–8]. Examples of implementation with various PUFs are summarized in Table 1.

2.2.1. Ring Oscillator PUFs

Ring oscillator (RO) PUFs are designed typically with 16 to 256 CMOS-based circuits, each oscillate at a slightly different value due to small variations during fabrication [9]. The pairing of two rings generates a consistent response, 0 or 1, if the first ring oscillates slower or faster than the second one. RO-based PUFs are widely used to secure field programable

gate array circuits (FPGA), and some integrated circuits. They can, however, be subject to side-channel analysis through electromagnetic interference. The set of addresses **A** generated from **T** can point to a set of RO pairs in a particular order. The set of instructions **I** point to a subset of ROs, and to a targeted value of the power supply. The responses of the PUF are the stream **K**.

### 2.2.2. Arbiter PUFs

Arbiter PUFs are designed with chains of the multiplexer (MUX) circuits, each allowing the transmission of electronic signals through two possible paths, up or down [10]. The chains feed Reset-Set latches, which switch to 0 or 1 when the delay at the Reset pad is either faster or slower than the delay at the Set. The stream of addresses **A** generated from **T**, point to a set of instructions that drive the MUXs. The set of instructions **I** point to a subset of instructions in order to generate the responses **K**.

### 2.2.3. SRAM-Based PUFs

Each cell of a static random-access memory (SRAM) is a flip-flop that has an equal chance to wake as a 0 or 1 after the occurrence of a power-off—power-on cycle [11–13]. Most of the cells tend to wake consistently, in the same way, thereby creating a fingerprint of the device. These types of PUFs are widely used because most electronic components already contain arrays of SRAM cells. The set of addresses **A** generated from **T** can point to a set of addresses in a particular order. The set of instructions **I** can point to a subset of the array. The responses of the PUF become the stream **K**. The entropy can be excellent when the SRAM array is large enough; however, methods to read the content of the memory are available when the device is lost to the opponent.

### 2.2.4. ReRAM-Based PUFs

This PUF is discussed in detail in this paper as it has interesting tamper-resistant features [14,15]. Each cell of a resistive random-access memory (ReRAM) has a unique resistance value that is compared to a median value. The value, 0 or 1, is generated from each cell depending on whether the resistance has been lower or higher than the median. The set of addresses **A** generated from **T** can point to a set of addresses in a particular order. The set of instructions **I** can point to a subset of the array. The responses of the PUF become the stream **K**. Other random access memory circuits such as DRAMs [16], Flash memories [17–19], and MRAMs [20,21] can also be used to design PUFs.

**Table 1.** Examples of challenge-response configurations for various PUFs.

| PUF | Challenges | | | Responses |
|---|---|---|---|---|
| | **T** | *IDAccess* | | |
| **RO** | To point at a set of M pairs of ROs | To point at a subset of N pairs of RO (N < M) | To avoid pairs oscillating at a similar frequency | Each N pair of ROs generates a 0 or 1 |
| **Arbiter** | M sets of instructions driving MUXs in the up or down position | To point at a subset of N instructions (N < M) | To avoid the sets of instructions known to be unstable | Each N set of instructions generates a 0 or 1 |
| **SRAM** | To point at M addresses in the SRAM array | To point at a subset of N addresses (N < M) | To avoid the SRAM cells known to be unstable | Each N cell of SRAM array generates a 0 or 1 |
| **ReRAM** | To point at M addresses in the ReRAM array | To point at a subset of N addresses (N < M) | To avoid the ReRAM cells known to be unstable | Each N cell of ReRAM array generates a 0 or 1 |

### 2.3. Use of Ternary States to Protect the One-Way Unclonable Functions

The addition of a third state allows the tracking of the portions of the PUF that should be masked because they are fuzzy, marginal, unstable, or fragile. The objective is then

to reduce BERs and improve reliability [22,23]. A thorough enrollment cycle to identify these "weak" portions and their masking during response generation, results in a better quality PUF, requiring little to no error-correcting scheme for cryptographic key generation. The knowledge of the addresses with fragile physical elements that can be damaged during normal operations have tamper-resistance properties [15]. A cryptographic protocol generating keys solely from the addresses of PUFs that are known to be reliable does not damage the weak portions of the PUF. However, an opponent trying to interact with the PUF without such knowledge could partially damage the PUF, leaving behind traces of the attack. A protocol to sense attacks could include reading these addresses at a very low electric current, to detect abnormal BERs.

Ternary-based logic, as opposed to mainstream binary logic, offers additional levels of security in cryptographic systems. The number of possible states is higher, which adds obfuscation and entropy. The addition of an additional state has been successfully integrated into PUF-based key generation schemes [23]. The handshake between a server initiating a new key generation cycle, and a client device equipped with a PUF, cannot be successfully initiated by a man-in-the-middle attack, which is unaware of the positions of the tri-states. Such a ternary representation can be combined with a scheme in which the keys generated by the addressable PUFs are different when the electrical currents injected into the physical elements change [15].

### 2.4. Error-Correcting Methods Versus Search Engines

Error-correcting schemes mitigate the differences between responses generated on-demand, and those collected during enrollment [24–28]. The responses of a PUF must be perfectly corrected in order to be used as cryptographic keys. The design of search engines is considered in this paper to replace error-correction methods, with the aim of enhancing tamper resistance.

#### 2.4.1. Error-Correcting Codes (ECC)

As shown in Figure 1, the challenges transmitted by the server (**T**, *IDAccess*) allow the generation of the response **K′** from the PUF, which should be similar to the initial responses **K** extracted from the information stored in a secure database by the server during enrollment. ECC is needed to uncover the stream **K** from the potentially erratic stream **K′**, and to generate error-free cryptographic keys. ECC exploits data streams called "helpers" to correct the erratic bits. In certain applications, the data helpers need to be protected by encryption schemes to avoid leakages to the opponents [29]. A fuzzy extractor embedded in the client device reads **K′** and the data helper to find **K**. The computing power consumed by the fuzzy extractor could be prohibitive for power-constrained IoTs, making these devices vulnerable to side-channel analysis. ECC has been successfully implemented with PUF BERs below 10%.

#### 2.4.2. Response-Based Cryptography (RBC)

RBC enhances the protection of the key generation process by eliminating the need to operate a fuzzy extractor at the client device level, see Figure 2. In lieu of generating a data helper from **K**, the client device hashes **K′** to generate the message digest **H(K′)**, which is transmitted to the server. The RBC is a search engine that is able to recover **K′** from **K** and the message digest **H(K′)** [30–32]. Through an iterative process, the RBC search engine hashes the initial response $K = K_0$ and compares the message digest $H(K_0)$ and **H(K′)**. If these two do not match, the RBC tests all data streams $K_{1k}$ from a Hamming distance of 1 of $K_0$. If these do not match, the RBC iterates and tests all data streams $K_{2k}$ from a Hamming distance of 2. Such an iterative process can handle Hamming distances up to 3, and 256-bit long responses. Beyond this, the latencies become prohibitive. A scheme fragmenting the data streams with nonces has been documented as effective with BERs as high as 20%. For example, if 256-bit long responses are impacted by 7% BERs, a fragmentation of 4 is applied; with SHA-128, **H(K′)** as a $4 \times 128$-bit long stream. Each fragment is filled with nonces and is hashed. The RBC does not necessarily consume less power than a mainstream ECC,

and it is not faster; however, the burden is moved to the server, thereby reducing both the vulnerability and the power consumption at the client device level. The hashing of **K'** is a light operation. Hashing methods such as SHA-128 or SHA-256 are extremely fast, and commercial implementation in hardware embedded in cryptographic processors is available. The RBC can also benefit from high-performance computing (HPC), graphic processor units (GPU), and associative processor units (APU), all of which leverage the possibility of distributing the search through parallel computations [33,34]. This approach can also leverage noise injection schemes relying on HPC solutions, which reduce the risk of exposure to certain attacks. The opponents need access to similar computing power in order to participate. The use of HPC is not recommended for the "RBC-light" version needed for key recovery, see Section 3.3.

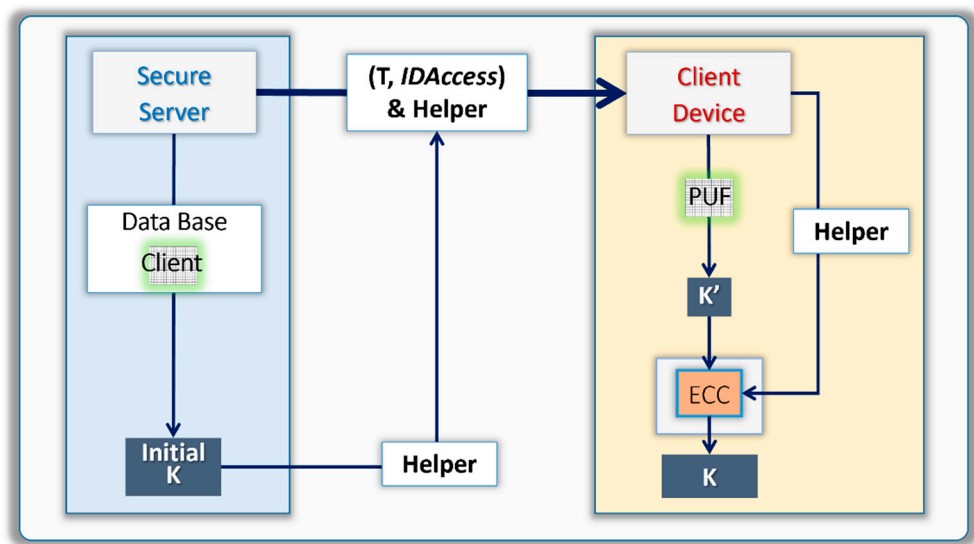

**Figure 1.** Block diagram of an ECC scheme. The challenges (**T**, *IDAccess*) generate the responses **K** from the image of the PUF stored by the server, and the response **K'** from the PUF. The helper transmitted by the server allows the ECC fuzzy extractor of the client device to recover **K** from **K'**.

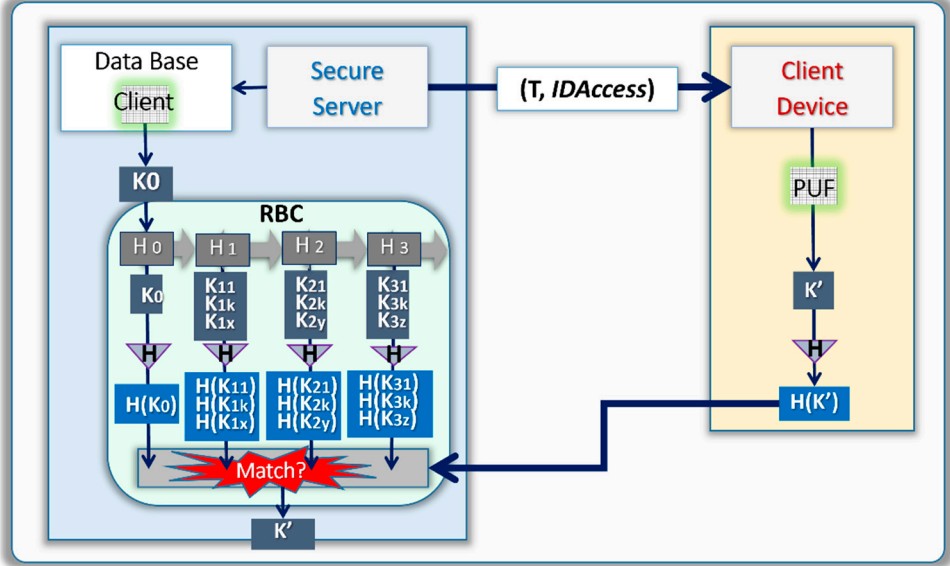

**Figure 2.** Block diagram of RBC search engine scheme. The challenges (**T**, *IDAccess*) generate the responses **K₀** from the image of the PUF, and the responses **K'** from the PUF. The RBC finds **K'** through an iterative process from **K** and **H(K')**.

## 3. Session Key Recovery with Ternary PUFs

### 3.1. Preparation Cycle—Session Key Encapsulation

The objective of this protocol, shown in Figure 3, is to protect a session key **Sk** by the embedded PUF during powering off cycles of a client device. It is assumed in this protocol that the databases of the client device could be lost to the opponent. Therefore, the information stored in these databases is considered public information. Without the PUF, or its image, the secret key **Sk** should not be retrievable. During the preparation phase, the challenges (**T**, *IDAccess*) generating the responses **K** from the database are transmitted to the client device by the secure server. To enhance security, it is assumed that the client device does not know the fuzzy positions that are tracked with ternary states. The data stream *IDAccess* allows the masking of the fuzzy positions, which has the objective of reducing BERs at the client level. This requires the thorough identification and storage in the database, by the secure server, of the erratic cells during an enrollment cycle. The session key **Sk** is then encrypted by the client device using the freshly generated responses **K'**. The information stored by the client device is (**T**, *IDAccess*) the challenges, **H(K')**, the hash message digest of the response **K'**, and **E(Sk, K')** is the ciphertext generated by encrypting the **Sk** with **K'**.

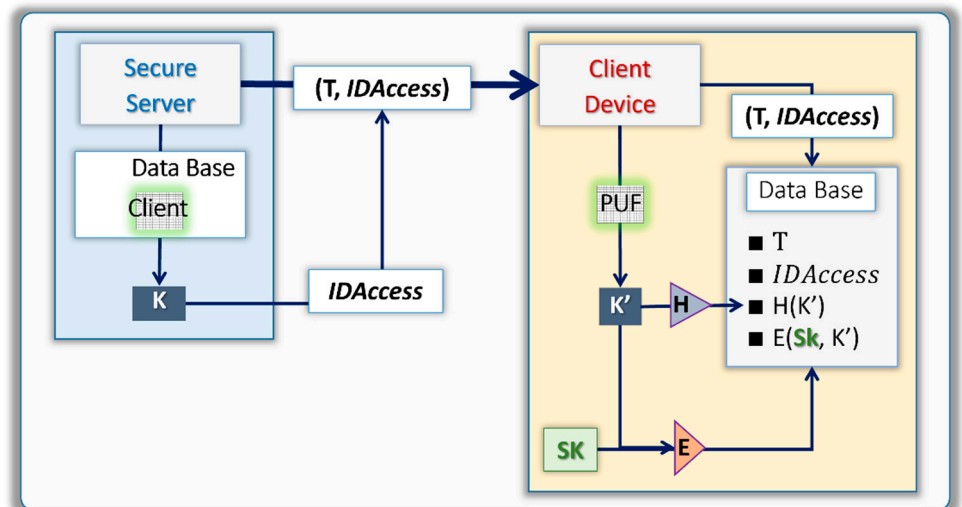

**Figure 3.** Block diagram of the encapsulation of the session key **Sk**. *IDAccess* is computed from the image of the PUF. The challenges (**T**, *IDAccess*) generate **K'** from the PUF. The client devices store the challenges, the ciphertext of **Sk** encrypted with **K'**, and the message digest **H(K')**.

### 3.2. Session Key Recovery

In the key recovery cycle, see Figure 4, the challenges (T, *IDAccess*) are retrieved by the client device, and used to generate the responses K″ from the PUFs that are not necessarily identical to **K'** due to the drifts of the parameters driving the PUF. The search engine, such as RBC, uses the message digest H(**K'**) to retrieve **K'** from K″.

Mainstream ECC schemes can replace the search engine in the architecture presented in Figure 4, whereby the message digest is replaced by a data helper. The session key **Sk** is recovered by decrypting the ciphertext with **K'**. The opponent cannot recover the session key without having access to the PUF, which can strengthen the system's tamper resistance. The session key can be used in various symmetrical cryptographic schemes such as AES and DES, or in public key infrastructures such as Elliptic curves, RSA, and Post Quantum Cryptographic codes such as Dilithium, Kyber, NTRU, Falcon, Saber, Rainbow, and Classic McEliece [35–42].

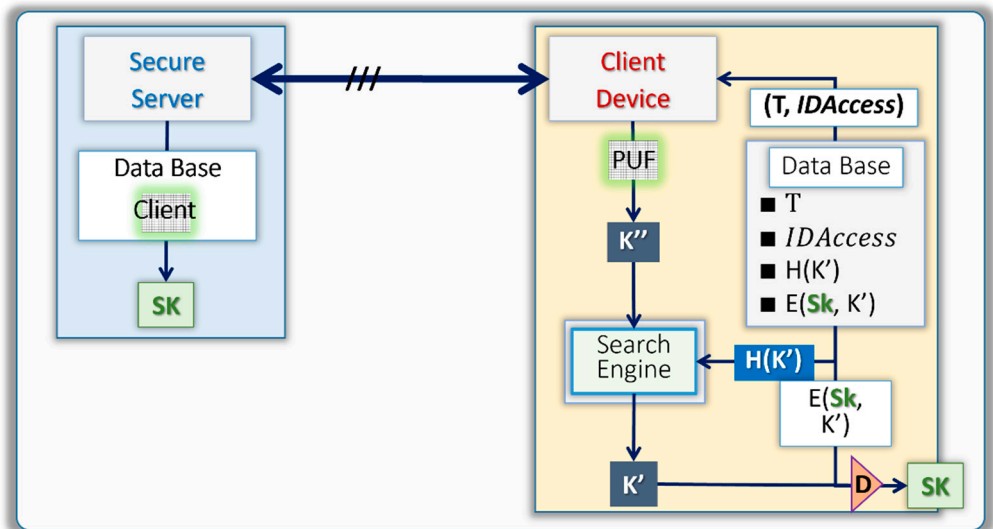

**Figure 4.** Block diagram of the recovery of the session key **Sk** by the client device. The challenges (**T**, *IDAccess*) generate **K″** from the PUF. The search engine finds **K′** through an iterative process from **K″** and **H(K′)**. The ciphertext is decrypted with **K′** to recover **Sk**.

### 3.3. Light Search Engine Implementation

A light version of the RBC was developed for the ReRAMs because the BERs were low enough and because the fuzzy positions of the PUFs were masked by the ternary states. In our model, the search was restricted to Hamming distances of 0 and 1. When no message digest matches **H(K′)**, rather than exploring higher Hamming distances, the process iterates through a new query of the PUF with the same challenges (**T**, *IDAccess*) to generate new responses. Using the numerical example of a case with average BERs of $2\,10^{-3}$ following a normal distribution, statistically, such a distribution creates an average of 0.5 bad bits on a 256-bit long stream. During on-demand response generation, such a PUF has approximately a 60% chance of experiencing zero bad bit, a 30% chance of experiencing one bad bit, an 8% chance of experiencing two bad bits, and a 2% of experiencing at least three bad bits. The RBC-light typically takes $10^{-4}$ s to check the Hamming distance of zero, 30 ms to check all configurations at a Hamming distance of one, 2 s to test all configurations at a Hamming distance of two, and 200 s to test all configurations at a Hamming distance of three. The latency to generate a fresh response from the PUF is only 10 ms, therefore re-setting the process after the search at a Hamming distance of one does cut the average latency. Statistically, 90% of the searches are positive after one query, and only 1% require a second query. A variation of the protocol was evaluated in which the client device has a database with a fuzzy state position that is able to find the full challenge (**T**, *IDAccess*) on its own, without assistance from the server. The security of such a scheme is slightly degraded as a third party could have access to the database containing these positions.

### 4. Content Protection with Ternary Unclonable Functions

One way to protect digital files is to encrypt them with a cryptographic key, such as the session key **Sk** presented in Section 3. The same session key can be used to either encrypt or decrypt multiple digital files. In this section, we present methods for delivering and protecting digital files, each of which is protected by its own set of keys. Rather than using on-demand challenge-response pairs (CRPs) to protect session keys, we suggest using a different CRP for each digital file. The random number **T** and the individual digital access parameter *IDAccess* of each CRP generates the response **K** protecting a particular digital file **M**.

### 4.1. Preparation Cycle—Encryption and Delivery of the Digital Files

As shown in Figure 5, the process of preparing the delivery of a digital file **M** starts with the generation of a set of challenges (**T**, *IDAccess*). **T** is obtained through a random number generator, and *IDAccess* is computed from the look-up table containing an image of the PUF. This allows for the masking of the ternary positions, as well as the parameters such as the electric current that is injected during the key generation. The digital file **M** is encrypted with responses **K**. An example of the protocol that encrypts and transmits the ciphertext to a client device is as follows:

- Both communicating parties have independent access to a shared password **PW**; number **T** and **PW** are XORed. The resulting stream is hashed with a SHA-3 generating **MD**, which is extended with a SHAKE to generate stream **A** for the m addresses:
  - $\text{MD} \leftarrow \text{SHA-3 } (\mathbf{T} \oplus \mathbf{PW})$
  - $\mathbf{A} \leftarrow \text{SHAKE}(\mathbf{MD})$
  - **A** is pointing at the m addresses of the PUF
- The m-bit long mask is retrieved from *IDAccess* to hide the addresses containing fuzzy positions. This leaves k positions, k < m, for response generation from the image of the PUF. The output is the k-long response **K**.
- The digital file **M** is encrypted into ciphertext **C** with **K**, the responses **K** are hashed with a SHA-3 to get **H(K)**, and the mask is XORed with **MD**.
- **T**, **C**, and **H(K)** are transmitted to the client device.

As presented in Section 2.1, it is important to prevent the opponent from knowing the fuzzy positions, as well as any other parameters. The suggested protocol assumes that **T**, **C**, and **H(K)** can be transmitted through non-secure channels. The information is protected by the password and needs *IDAccess*, as well as the PUF, or its image to disclose the digital file **M**. At this point of the protocol, the client device does not have access to the digital file. As needed, the server can prepare multiple digital files with multiple challenges. In the commercial context and example of the delivery of movies, the service provider can send a stream of files to its customer, each encrypted with its own responses.

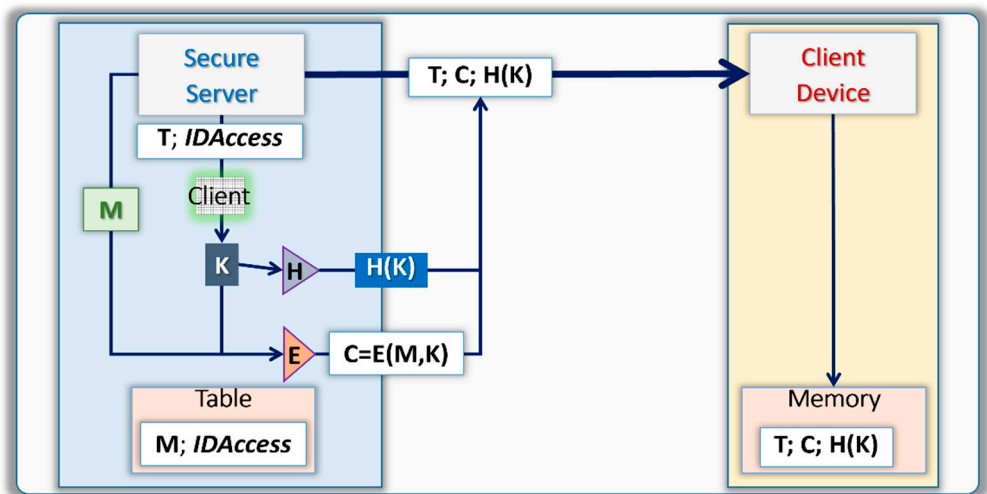

**Figure 5.** Block diagram of the encryption of digital file M. The challenges (**T**, *IDAccess*) generate **K** from the image of the PUF. **M** is encrypted with **K**. The client device stores **T**, ciphertext C, and message digest **H(K)**. The server keeps *IDAccess* in a look-up table.

### 4.2. Decryption of the Digital Files by the Client Device

To trigger the read cycle, the server communicates the missing piece of the challenges, *IDAccess*, which the client device combines with **T**, see Figure 6. The one-way unclonable function generates the responses **K'** from the PUF, providing the necessary information to the client device to generate the key **K** from $\mathbf{K'} \leftarrow \Psi (\mathbf{T}, \textit{IDAccess})$.

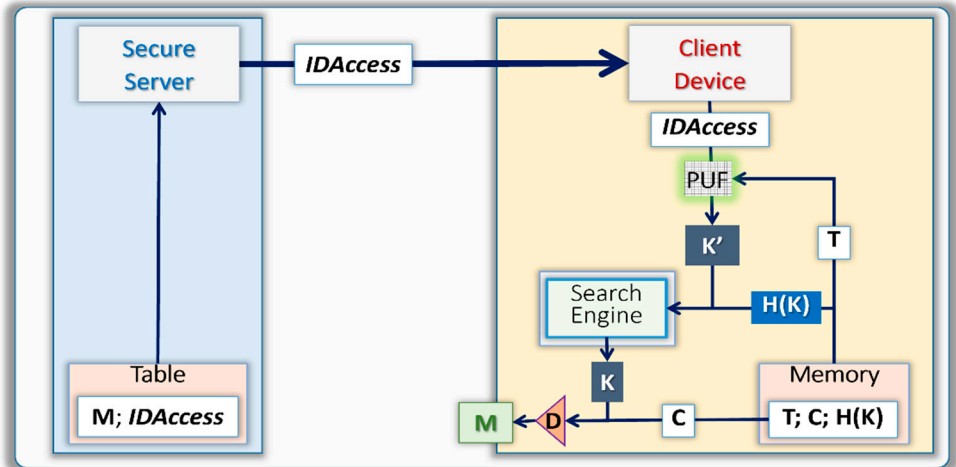

**Figure 6.** Block diagram of the decryption of **M**. The server transmits *IDAccess* to trigger the process. The challenges (**T**, *IDAccess*) generate **K'** from the PUF. The search engine finds **K** through an iterative process from **K'** and **H(K)**. The ciphertext is decrypted with **K** to recover **M**.

The search engine, similar to the one described in Section 3.2 for session key recovery, recovers the initial responses **K** from **K'** and the message digest **H(K)**. This allows access to the digital file **M** by decrypting the ciphertext **C** = **E(M,K)**. Any symmetrical encryption scheme such as AES or DES can be considered for use in this method. We used AES-256 in this study. A commercial application example is where a service provider communicates *IDAccess* after receiving a payment from a client's device for a digital file such as a movie. Other practical examples are suggested in the summary section of this paper (Section 7).

### 4.3. Protection of Digital Files Stored by IoT Terminals

The method proposed in this section to protect digital files is a variation of that presented above. Here, the information generated at the IoT level and which is stored by the IoT for future use is protected. The objective is to prevent a third party from simply retrieving information from the memory unit of the IoT, which is equipped with its own one-way unclonable function. During the preparation cycle, as shown in Figure 7, the server generates the challenges (**T**, *IDAccess*) from the image of the PUF. The client device goes through the following steps:

- Retrieves the challenges (**T**, *IDAccess*).
- The responses **K** are generated with the PUF, from the challenges.
- The IoT hashes **K** for the search engine.
- The file **M** is encrypted with **K** to generate the ciphertext **C**.
- The IoT stores **T**, **C**, and the message digest **H(K)**, but not *IDAccess*, which is only stored for future reference by the server.

The IoT can only decrypt message **M** after receiving from the server the individual digital access information *IDAccess*. The server can delegate the responsibility to dispatch *IDAccess* to a third party. If the one-way function is tamper-resistant, an opponent cannot decrypt **M** without an image of the one-way unclonable function. An example of the implementation of the decryption of **M** by the client device is as follows:

- Receive *IDAccess*.
- Read from the memory number **T**, ciphertext **C**, and message digest **H(K)**.
- Generation of **K'** from the one-way unclonable function.
- Retrieve **K** from **K'** and **H(K)** with a search engine.
- Decrypt the digital file using **K** as a cryptographic key.

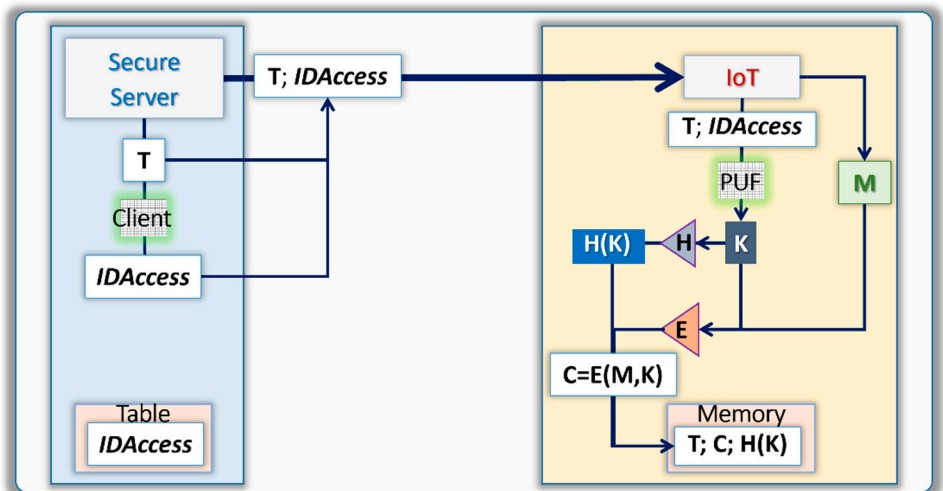

**Figure 7.** Block diagram of the encryption of file M for IoTs. The server prepares the challenges (**T**, *IDAccess*). The digital file **M** is encrypted with **K** by the IoT, which stores only **T**, ciphertext **C**, and message digest **H**(**K**). The server keeps *IDAccess* in a look-up table for future operations.

## 5. Implementation with SRAM and ReRAM Devices

The methods proposed to recover session keys and to protect digital files can be implemented with any one-way function that verifies Equation (1) in Section 2. The implementation with SRAM is straightforward, as the devices are commercially available. The implementation with pre-formed ReRAM requires the design of custom circuits, as well as a rather complex data acquisition board, that interfaces with the Wi-FIRE ChipKit engineering board from Digilent. One of the aims of such a dual implementation was to benchmark the tamper-resistant ReRAM-based solution with more established SRAM-based solutions. The core engine of the engineering board is a microcontroller manufactured by Microchip with a 200 MHz 32-bit MIPS processor, a 2 MB embedded Flash, and a 500 KB SRAM. The board is powered via a USB port, and it operates at 3.3 volts.

As shown in Figure 8, the custom boards containing the memory devices are plugged into the ChipKit board. In this analysis, the engineering boards communicated with the PC through a USB cable. To collect the initial responses, the SRAM and ReRAM systems went through quick enrollment cycles, where the unstable cells were categorized with a third state; all initial responses were stored in look-up tables. The server generates 256-bit long keys from the look-up tables, and the Chipkits generate 256-bit long keys from the PUFs. The SRAM and the ReRAM systems use the same RBC and also similar handshake schemes between the server and the client devices. Different protocols are needed to generate the keys from each PUF, as described below.

### 5.1. Description and Analysis of the SRAM Implementation

A set of switches allows quick power-off cycling of the SRAM to reset the device prior to the response generation. The load around the I/Os of the SRAM are such that a power-off cycle still takes at least two seconds before the flip-flops of each cell are properly grounded. The read cycles of the SRAM are extremely fast, a 256-bit long key only needs 10 μs. Most SRAM cells always wake in the same state, as a "0" or a "1" after power off—power on cycles; however, 3 to 5 % of the array changes states in each cycle. This could result in high BERs, so during the enrollment cycle of each SRAM, we performed repetitive power-off-on cycles to identify the unstable cells.

Examples of the experimental results are shown in Figure 9. The results represent 30 successive key generation cycles, lasting 2 s each; the enrollment had 100 cycles at room temperature. At each key generation cycle, the server randomly selects 512 positions, of which 256 are masked to keep only the most stable cells. The 256-bit long keys generated from the look-up table are on the left, those from the SRAM-PUF are on the right, and are

followed by the count of errors. The average BERs were about one percent, i.e., 3 erratic bits per key, which were due to both the SRAM instability and noises in the electronic system. Such BERs are well within the capability of the RBC, with false reject rates (FRR) below 1%, and one-second average latencies. After about 100 cycles, we were left with approximately 88 % of the cells waking in the same state, the rest of the population had at least one erratic response. At between 100 cycles and 1000 cycles, less than 5% of the additional cell population was still unstable.

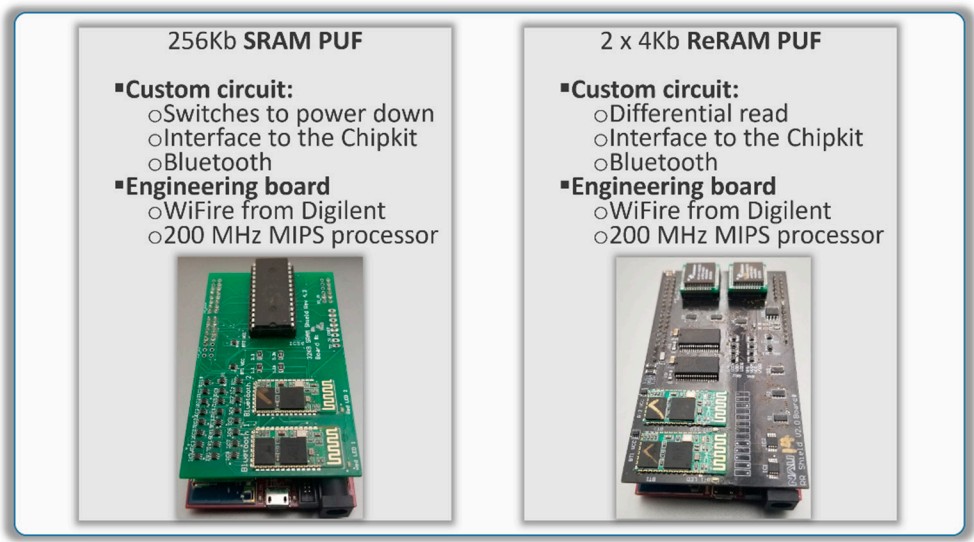

**Figure 8.** Pictures of the hardware set up for the SRAM (**left**), and the ReRAM (**right**). The data acquisition board for the memory chips is plugged into "WiFire" engineering boards provided by Digilent. A pair of ReRAM devices are needed for the differential circuit generating responses.

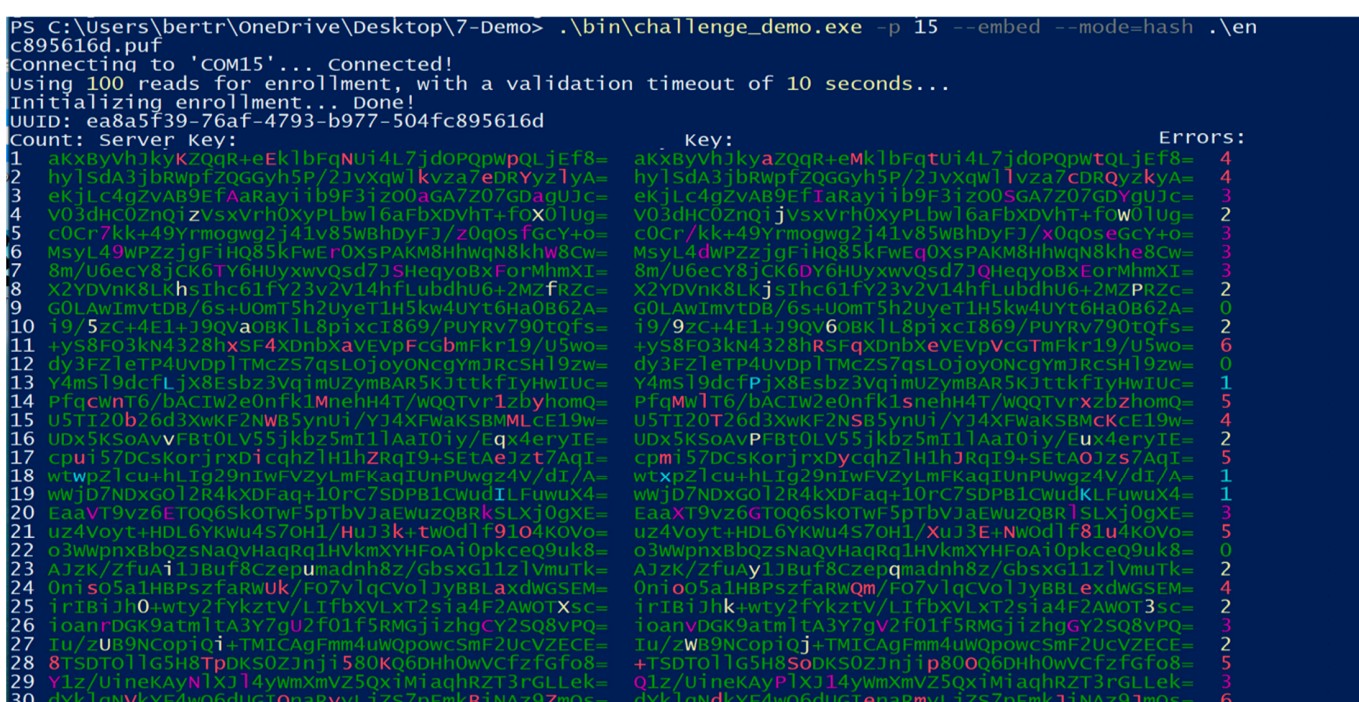

**Figure 9.** Set of thirty 256-bit long key generation cycles with SRAM. The keys generated by the server from the look-up table are on the left, those generated from the SRAM PUF are on the right, followed by the count of errors. BERs are in the one percent range.

The results presented below in Figure 10 were derived from using the same methodology of varying the number of enrollment cycles at different temperatures. We performed tens of thousands of key generation cycles to obtain statistically significant results, which is necessary to report BERs in the one part per million range. The SRAM was subjected to 1000 enrollment cycles at different temperatures: 0 °C in light blue, 20 °C in red, 40 °C in grey, 60 °C in orange, and 80 °C in dark blue. The BER was then computed with responses generated in the 0 °C to 80 °C temperature range for each temperature of enrollment. We observed smaller BERs, in the $2 \times 10^{-5}$ range, when the responses were generated at the same temperature as the enrollment; however, the BERs quickly degraded when these temperatures did not match. A multi-temperature enrollment, shown in green in Figure 9, yielded BERs in the $1 \times 10^{-5}$ to $2 \times 10^{-6}$ range. However, such a high-quality enrollment is time-consuming as we needed to pause two seconds between cycles, in addition to the 3 s needed to test the devices. The multi-temperature enrollment took $5 \times 5000$ s, or 7 h, which is not practical for some applications.

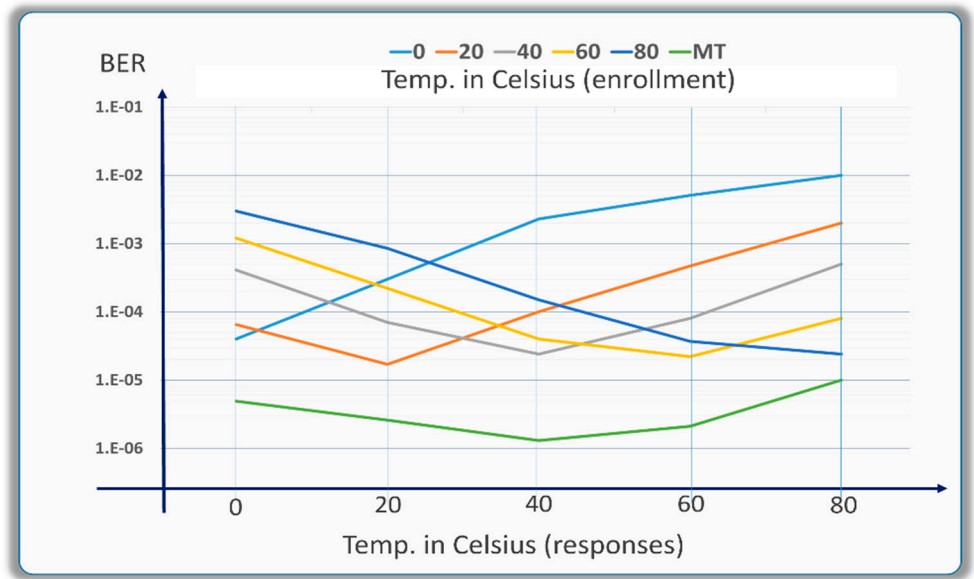

**Figure 10.** BER of the SRAM PUF (*Y*-axis). The enrollment consisted of 1000 power off/on cycles at 0 °C, 20 °C, 40 °C, 60 °C, and 80 °C. The responses were generated at 0 °C, 20 °C, 40 °C, 60 °C, and 80 °C (*X*-axis), for each temperature of enrollment. Shown in green, the enrollment was conducted at multiple temperatures, and the BERs were low regardless of temperature during response generation.

*5.2. Description and Analysis of the ReRAM Implementation*

The two ReRAM circuits were designed with 1.2 volt I/Os, which require voltage shifters and pin expenders to be connected to the ChipKit. Such an interface is complex and slows down the engineering board. We anticipate that the final iteration of this technology will be based on 3.3 volt I/Os. The decision to design the board with two 4 Kbit ReRAM circuits, rather than one, allowed faster read cycles of about 10 ms for a 256-bit long key, and higher tamper resistance. To increase entropy, the responses were generated by injecting a small electric current that can randomly vary from 100 nA and 800 nA every 100 nA, in pairs of cells, each located in separate arrays of 4 k cells. The number of possible challenge-response pairs in this experiment was $8 \times 4096 \times 4096 = 128$ million. The response was a "0" when the first cell had a resistance lower than the second cell, and a "1", in the opposite configuration.

To reduce BERs, the cells having similar resistance values should be removed. Examples of the experimental results, leveraging the differential pre-formed ReRAM-based system, are shown in Figure 11. These results represent 30 successive key generation cycles, lasting 2 s each, and after this, there was a quick enrollment of both ReRAM arrays at room temperature for 80 read cycles.

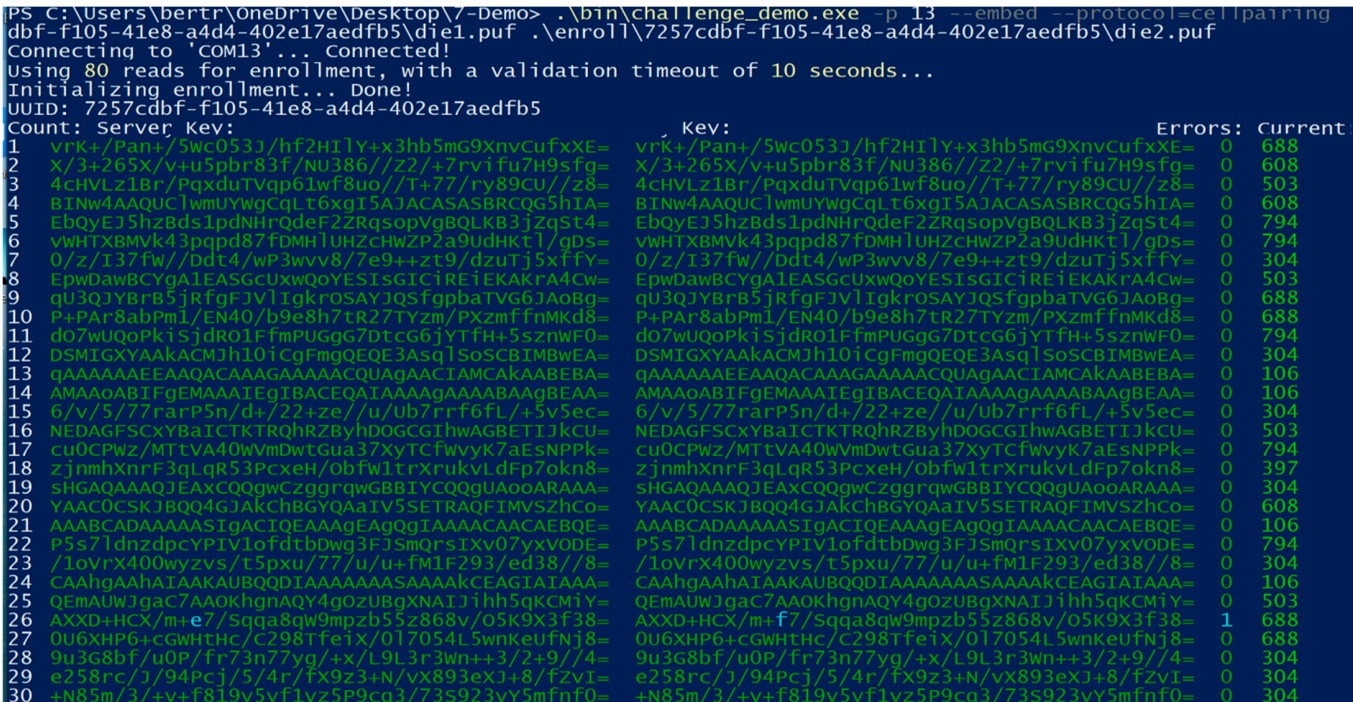

**Figure 11.** Set of thirty, 256-bit long key generation cycles, with two ReRAM devices. The keys generated by the server from the look-up table are on the left, those generated from the ReRAM PUF are on the right, followed by the count of errors. The BERs are very low, in the $10^{-4}$ range.

The starting point for the protocol is that 512 pairs of cells are randomly selected, of which 256 pairs act as a buffer, keeping the remaining 256 pairs further apart in resistance values. The 256-bit long keys generated from the look-up table are on the left, those from the ReRAM-PUF are on the right, followed by the count of errors. Here, we only observed one error, during the key generation cycle number 26, which could be due to the instability of the ReRAM or noise in the electronic system. Such BERs are within the capability of a light version of the RBC, using keys located at a Hamming distance of zero or one.

In Figure 12, an experiment to quantify the effect of the size of the buffer is presented. To generate 256-bit long keys, 256 + **k** pairs were selected, with **k** being the number of extra pairs, with the lowest differences in resistance values, that were removed. The goal was to remove enough pairs to generate 256-bit keys with BERs in the part per million (ppm) range. The size of the buffer **k** can be lowered when the cell-to-cell differences in resistance values are large enough, and predictable, compared with fluctuations due to the measurement scheme. In our experiment, we designed circuits with high accuracy in the differential read, with latencies below 100 μs per read-cycle. The lowest buffer sizes, around 29 pairs, were observed at room temperature and 800 nA. The largest buffer sizes, around 35 pairs, were observed at 80 °C and 100 nA.

The protocols presented in the experimental section of this paper used 256 pairs as a buffer, which was anticipated to generate BERs way below 1 ppm. Many other parameters besides the ReRAM PUFs impacts BERs at such low levels; therefore, the exact quantification of the BERs is not easy and will be the subject of future research.

The differential protocol we developed for the two ReRAM circuits also protects the system against an opponent trying to read the resistance values of the ReRAM arrays. After enrolment, the two circuits can be mounted in such a way that only differential measurements can be conducted. In Figure 13, the number of days needed to read all pairs, as a function of the size of the array, is shown, together with the number of possible currents injected into each cell. For example, it takes 1000 days to read all possible pairs produced by two arrays of 512 K-bits each, and there are 25 possible levels of injection currents.

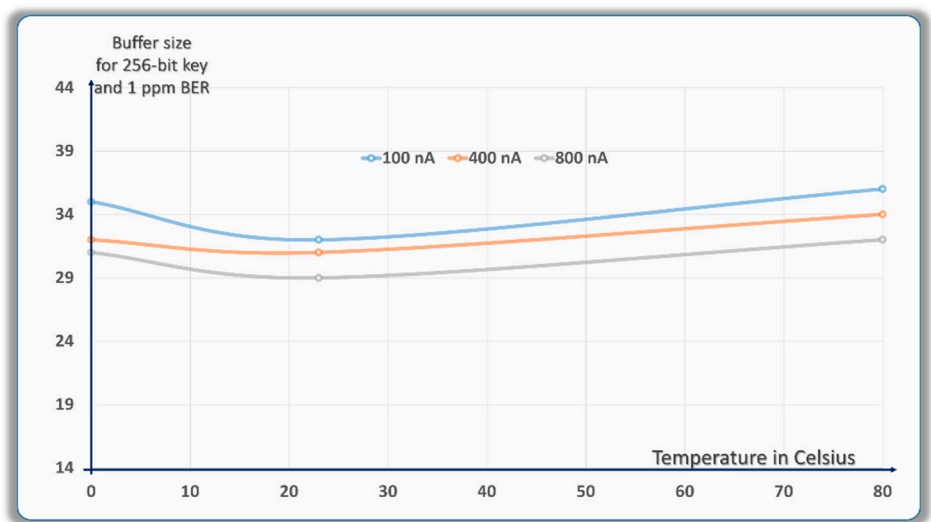

**Figure 12.** The size of the buffer (*Y*-axis) that is required to produce a BER at 1 ppm for 256-bit long keys generated from a pair of ReRAM circuits. The lowest buffer is observed with currents of 800 nA, from 0 °C to 80 °C (*X*-axis). In this case, 288 cells are required to produce a 256-bit long key with BER at 1 ppm.

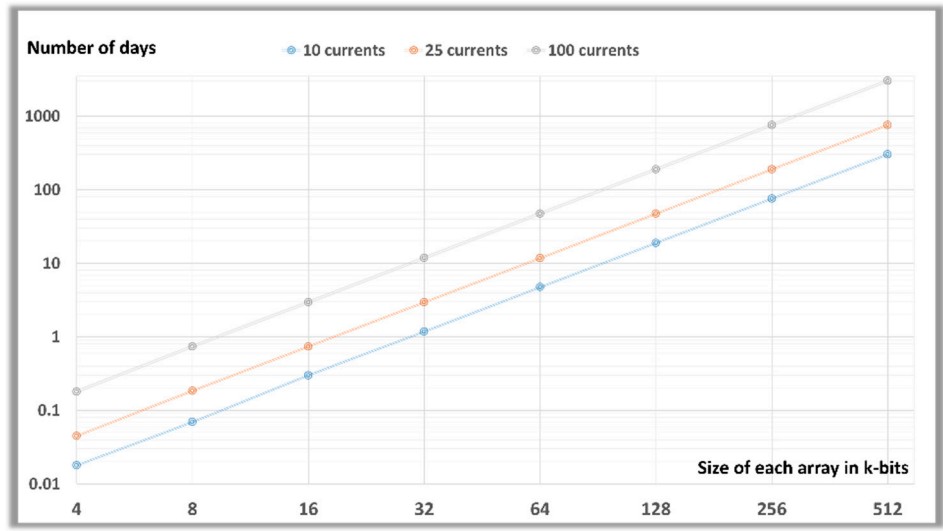

**Figure 13.** Modelling the time in days to complete the crypto-analysis of pairs of ReRAM devices. The time to completion increases exponentially, proportional to the size of the array and to the number of possible injected currents.

### 5.3. Comparative Analysis of SRAM versus ReRAM Schemes

SRAM PUFs are widely available, cheap, easy to use, and fast [43]. This is a perfect technology to use to implement the methods presented in this paper, namely, key recovery, content delivery, and digital file protection. SRAM PUFs are a perfect fit for a range of applications that are not directly exposed to side-channel attacks, and applications where the risk of the device being infiltrated by an opponent, is low. Examples of areas of application include the protections of the IoTs at home, in the office, and in other areas that are relatively safe. One of the tradeoffs of using SRAM PUFs is the additional cost of enrollment that is needed to operate at low BER. The repetitive power off/on cycles are time-consuming. A remedy for this is to use powerful error-correcting schemes and to accept higher BERs. The ReRAM technology is not yet as pervasive as SRAMs; however, its potential to design tamper-resistant solutions is observable, particularly in the pre-forming

range. A comparison between the two technologies, based on the analysis presented in this section, is summarized in Table 2.

- Entropy: number of cells or pairs: The entropy of SRAM-based cryptosystems is proportional to the size of the array; however, the cost of enrollment also increases at the same rate. The differential protocol comparing the resistance value between the cells belonging to small 4 Kb ReRAM arrays involves 16 million possible pairs. With 8 different levels of possible currents, as tested in this study, the number of possible pairs reach 126 M. This number is scaled linearly by increasing the number of levels of current and with the square of the size of the array.

- Bit error rates of the responses: The BERs of SRAM PUFs are reduced by increasing the number of power off/on cycles at different temperatures. As shown in Figure 10, BERs in the $2\ 10^{-6}$ range is possible at a cost of enrollment cycles lasting multiple hours, which lacks practicality. Conversely, the way to reduce the BERs of two ReRAM arrays, driven by the differential protocol, is to increase the size of the buffer, which does not require longer enrollment times. Considering the difficulty in quantifying extremely low BERs, an extrapolation of the data reported in Figure 12, points to BERs in the $1\ 10^{-8}$ range, with buffer sizes large enough and with the appropriate screening of unstable cells.

- Enrollment cycles: One of the values of the differential protocol is to cut the enrollment time. There is no need to test the pairs of ReRAM cells upfront during enrollment, testing each array thoroughly is enough to generate the initial response from a look-up table. In the analysis performed in this study, eight thousand cells were tested during the enrollment of 15 min, rather than the 128 million possible pairs. The measurement of the resistance of a cell is analog; therefore, unlike reading an SRAM cell, there is no need to repeat the measurements to quantify the proportion of "0" or "1".

- Response cycles: Generating responses from the SRAM PUF is extremely fast after powering on the device. Minimizing latencies of the response generation of pre-formed ReRAM PUF has been a challenging task due to the high resistance values that could reach 10 MΩ. In the differential protocol, there is no need to measure these values, the only information needed is to find which cell has the higher resistance value of the two. This allows for an optimization of the circuitry. In this study, we found that 10 ms are enough to read 256-bit long streams. Further reductions in latencies have a negative impact on the BERs, as the measurement becomes noisy.

- Crypto-analysis: One possible attack, which is a major problem for certain applications, is when the terminal device is under the control of the opponent for even a short period of time. In this instance, it is possible to read the SRAM in a matter of seconds after power off/on cycles. The bulk of the information needed for key generation can be recovered after 100 cycles, which takes about 5 min. Pairs of ReRAM cells are more difficult to attack. The two 4 Kb ReRAM arrays are tested separately, upfront, during quick enrollment cycles. The circuitry for the ReRAM PUFs is such that when the two arrays are mounted on the custom board, the user only has access to differential reads, without having access to the individual devices. Therefore, a crypto-analysis requires that 128 million pairs be read, which takes about 4.4 h. As shown in Figure 13, two 512 Kbit arrays take 9 years to be differentially read.

Pre-formed ReRAMs have additional physical properties that also enhance tamper resistance:

- Ability to sense attacks: The design of sensing elements inserted in the ReRAM arrays operating in the pre-forming range has been reported [15]. An opponent exploring the ReRAM arrays without knowledge of the vulnerable cell population has a high probability of damaging these cells. The cryptosystems developed in this study avoid this population; therefore, it is possible to monitor the potential infiltration of a crypto-analyst and to detect an attack.

- Self-destruct mode at low power: ReRAMs are designed to operate in the set/reset mode after the forming operation. The forming operation in a ReRAM is a non-reversible process that usually starts with voltage stress in the 1.5-volt range. In case of an attack, the user can trigger a self-destruct mode of the ReRAM cells by initiating the forming cycles. Only partial cycles are needed, as the objective is to form enough cells to make the PUF useless, for example, half of the cell population.
- Radiation hardness: SRAMs are vulnerable to ionizing radiation; however, in this particular application there is a mitigation process of performing power off/on cycles before each response generation. The likelihood of several cells being impacted by radiation just before response generation cycles is small; therefore, the impact on the BER is anticipated to be limited. The ReRAM technology is known for being rad-hard [44]. The 4 Kb arrays in this study were manufactured with the conductive bridge RAM (CBRAM) technology that has been tested as more stable under ionizing radiation than the more traditional ReRAM technology that can be impacted by migrations of oxygen vacancies.

**Table 2.** Comparison between SRAM and pre-formed ReRAM based key generation protocols. The enrollment cycles of the SRAM are very slow. The cell pairing scheme driving the ReRAMs at different currents increases entropy and enhances tamper resistance.

| Factor | 256 Kb SRAM | 2 × 4 Kb ReRAM |
|---|---|---|
| Commercial availability | Broad | Limited |
| Entropy: number of cells/pairs | 256 k cells | 128 M pairs |
| BER responses | $2 \times 10^{-6}$ | $1 \times 10^{-8}$ |
| Latencies: enrollment cycle | 7 h | 15 min |
| Latencies: responses/256 bits | 10 μs | 10 ms |
| Crypto-analysis | 5 min | 4.4 h |
| Sense attack | No | Yes |
| Self-destroy | No | With 1.5 V |
| Radiation hardness | Limited | Yes |

The SRAM technology for the design of outstanding PUF-based cryptosystems has been demonstrated by industrial suppliers and academic institutions. It is not the objective of this analysis to criticize SRAMs in favor of largely unproven technology. The authors acknowledge that it will take years of research before ReRAM technology reaches similar levels.

## 6. Characterizing the Key Recovery from ReRAM PUFs

### 6.1. Rates of Erratic Keys Recovered from ReRAM PUFs

One of the explanations for the low BERs reported in Section 5 is the enrollment cycles which populated the look-up tables with quality information on the PUFs. The PUFs were subjected to 80 successive reads in order to identify, then minimize, the impact of the noisy reading cycles. This resulted in a reduction of the differences between the responses generated from the PUFs and the original responses that were stored in the look-up tables. The key recovery protocols are much more challenging. They can face higher BERs as the initial responses are the result of a single read, which could be noisy. Therefore, we conducted a new analysis to quantify the BERs relevant to the key recovery protocols. See Figure 14, in which the PUFs are used twice.

This figure shows examples of 44 key recovery cycles; the left column displays the keys generated during the first read from the PUFs and the right column shows the keys generated from the same PUFs during the recovery cycles. The numbers of errors between the two are shown in the last column. As was performed in Section 5, two pre-

formed ReRAM chips were used to generate 256-bit long keys using the differential pairing protocol. The resultant BERs are higher than those reported in Section 5; however, they are still relatively low, and are well within the RBC-light capabilities.

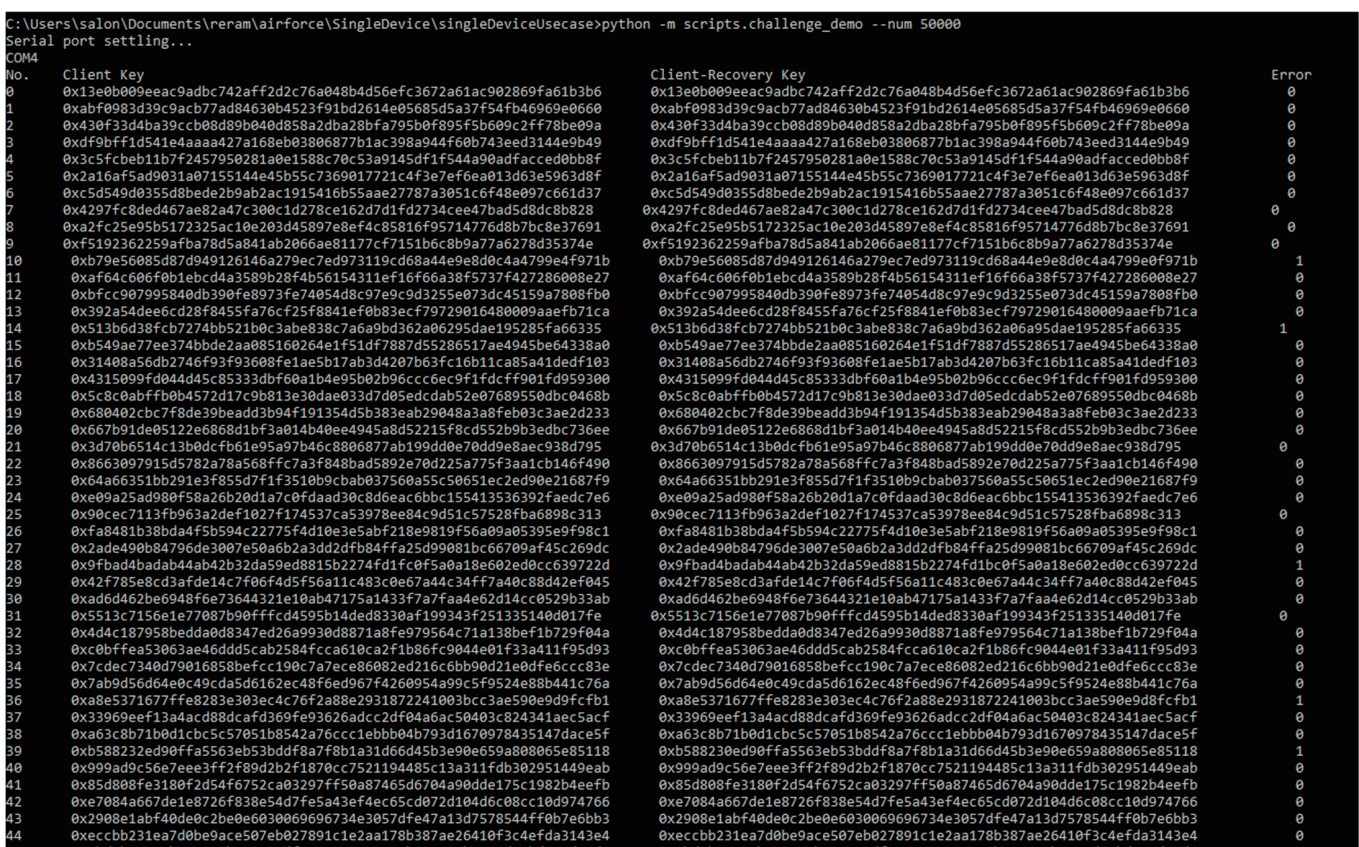

**Figure 14.** Forty-four 256-bit long key generation cycles. The keys on the left were initially generated from the ReRAM PUFs. The keys on the right were generated from the same addresses during recovery cycles. The number of mismatching bits between keys is counted in the right column.

To quantify the BERs, we performed 5000 successive cycles similar to that presented in Figure 14. The results are summarized in Figure 15, which shows on a log scale the probability of the occurrence of erratic keys as a function of the numbers of errors for 256-bit long responses. The average BERs observed in this experiment were 6.148 $10^{-4}$. These BERs created 787 bit-errors out of the 5000 pairs of 256-bit long keys. Such error rates are even lower than those given as examples in Section 3.3, in which the use of the RBC-light is suggested. The distribution of erratic keys observed here is well described by a Poisson distribution having a parameter λ equal to the average number of erratic bits per 256-bit long key:

$$\lambda = 256 \times \text{BER} = 256 \times 6.148 \times 10^{-4} = 0.1574 \tag{5}$$

- 4290/5002 keys (85.8%) have zero errors versus a Poisson distribution at 85.4%
- 643/5000 keys (12.9%) have one error versus a Poisson distribution at 13.4%
- 58/5000 keys (1.2%) have two errors versus a Poisson distribution at 1.06%
- 8/5000 keys (0.16%) have three errors versus a Poisson distribution at 0.06%
- 1/5000 keys (0.02%) have four errors versus a Poisson distribution at 0.002%

The RBC-light that searches for erratic keys with a Hamming distance of no more than one is expected to find the matching keys in 98.5% of cases. The need to generate a second response occurs in only 1.5% of the cases, and a third response in 0.02% of cases. False reject rates (FRR) of the key recovery will occur if the latencies of the RBC are too long due to an excessive number of iterations, and lengthy PUF response generation cycles.

Further optimization of the BERs can be achieved by increasing the enrollment cycles and identifying more cells that are unstable. Another opportunity for optimization is to impose higher differences in resistance between the selected pairs of cells.

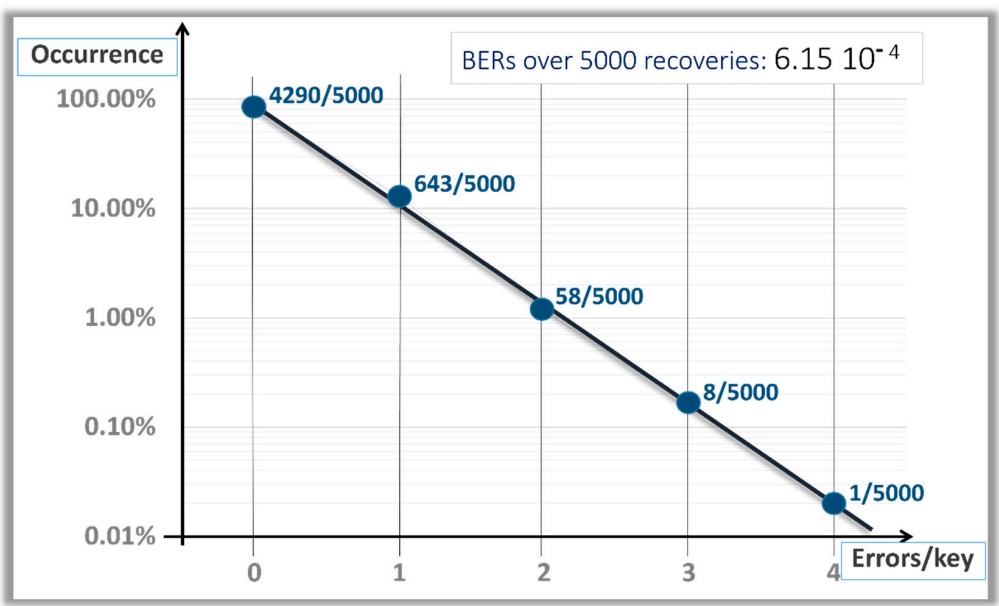

**Figure 15.** Plotting the occurrence of erratic 256-bit long keys generated from the ReRAM PUFs. Out of 5000 key recovery cycles, 4290 keys had zero errors, 643 keys had one error, 58 had two errors, eight had three errors, and one key had four errors.

### 6.2. Latencies for the Key Recovery Protocols with ReRAM PUFs

As part of the experiment presented above, we completed the full key recovery protocol, including the RBC-light. When the Hamming distance exceeded one, an additional key generation cycle from the PUF was performed to enable the recovery of the initial 256-bit long keys. A total of 5000 key recovery cycles were performed to quantify latencies. The results are as follows, as shown in Figure 16:

- The average latency to recover 4290 keys without error is 2.11 s, which is mainly due to the time it takes to read the 256 addresses from the pre-formed PUFs.
- The average latency to recover 643 keys with one error is 2.56 s. This includes an additional 40 ms for the RBC-light.
- The average latency to recover 58 keys with two errors is 7.3 s. The additional delays are due to the need to read the PUFs several times.
- The average latency to recover 8 keys with three errors rose to 10.6 s for the same reason.
- The average latency to recover the last key with four errors was more difficult and took 35.1 s. The difficulty here was the necessity to handle several cells that had responses that always differed from the initial response. We suspect that the initial read was noisy. This type of problem can be resolved by reading the key, multiple times, during the initial cycle and erasing the bad ones. However, in most use cases, a latency of 35 s every 5000 cycles is perfectly acceptable.

Despite the fact that the hardware that we designed for this study is far from being optimized in terms of stability, noise, and latency, the overall performance of the key recovery schemes is in line with the study objectives. With this protocol, 100% of the 5000 keys were recovered, with an average latency of 2.25 s. Considering that the RBC-light, with a Hamming distance of one, takes less than 40 ms, the bulk of the delays were due to the response generated by the PUF, which takes about 2.0 s.

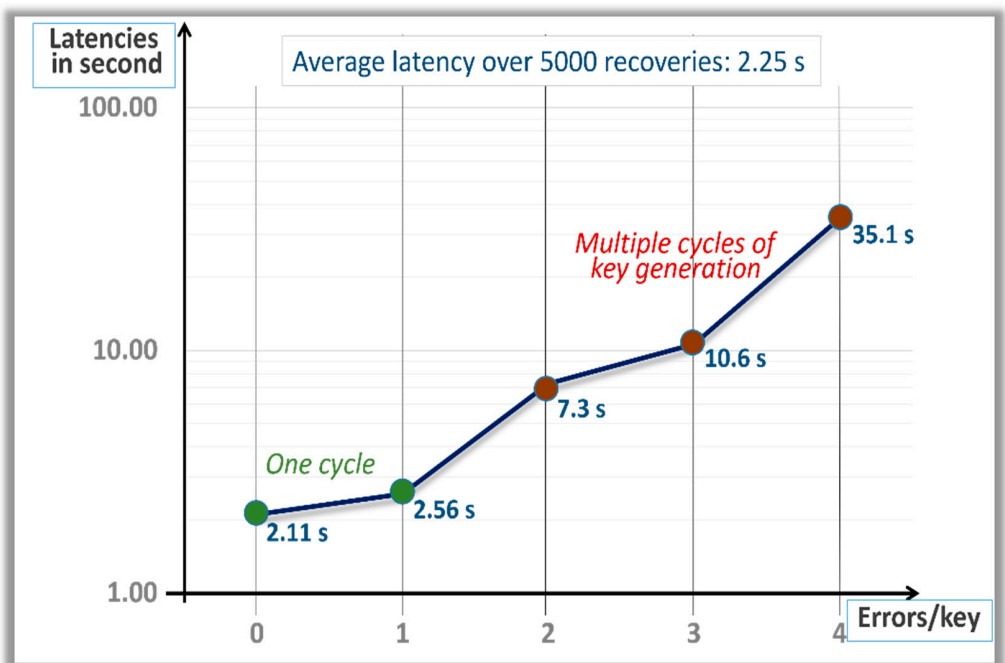

**Figure 16.** Plotting the latencies for key recovery in seconds from the ReRAM PUFs. When the number of erratic bits is zero or one, the RBC-light can find the matching key in one cycle, within 3 s. Multiple cycles of key generation from the PUF are needed at higher error rates.

*6.3. Software and Security Considerations*

The development of the protocols presented in this section was mainly based on generic software, that can eventually be implemented in hardware with secure components, as follows:

- The XOR functions concatenated the input parameters, such as random numbers, with passwords for multi-factor access control.
- The hash function SHA-3 (512 bit), with its one-wayness, is at the core of several layers of protection, including:
  - To convert the XORed input parameters into the message digest MD that feeds the XOF and selects the addresses of the PUF used for response generation. MD is also used to protect IDAccess after XORing operations.
  - As part of the RBC, the hashing of the responses is used to uncover the original responses.
- The XOF SHAKE converted the MD into the set of addresses pointing to the PUF.
- The session keys were encrypted using AES-256, and the keys were generated from the original responses of the PUF. They were decrypted with the keys retrieved from the RBC-light, and with the fresh responses from the same PUF after RBC correction.
- The session keys generated from the pre-formed ReRAM PUFs were tested successfully as seed data to generate private and public key pairs for elliptic curve cryptography, Dilithium learning with error cryptography [37], and Saber learning with rounding cryptography [39].

The objective of this research was not to develop a final product, which has to mitigate a potential list of attacks. For example, all custom software, including the RBC, was not designed to prevent side-channel analysis. The generic software modules, i.e., XOR, SHA-3, SHAKE, and AES was written in C and downloaded in the 200 MHz RISC microcontroller, rather than being executed through a secure crypto-processor. The aim was to focus on the development of an efficient key recovery scheme with acceptable performance, not to tackle and solve all security issues.

### 7. Summary and Future Research

The analysis reported in this paper show that the proposed ReRAM based solutions outperform SRAM PUF-based schemes in terms of BERs and tamper resistance. Bit error rates below the $10^{-3}$ range were demonstrated in the keys generated from ReRAMs operating in the pre-forming range with the differential cell pairing protocol. Unlike with SRAMs, such low BERs do not necessitate lengthy enrollments to remove the unstable cells. The differential protocol keeps only those pairs of cells with resistances further apart from each other. The BERs are reduced when the proportion of pairs that are masked by the protocol is large enough. Low BERs enable the use of a small search engine, the RBC-light, as a replacement for power-consuming error-correcting schemes, without fuzzy extraction, and without data helpers. The combination of tamper resistance, low BERs, and low power correcting methods facilitated the development of an end-to-end cryptographic system to deliver and protect digital files.

Future work: Unlike SRAM PUF based solutions, which are commercially available, and which have solid performances, the deployment of schemes using pre-formed ReRAMs requires additional research, that is not underestimated by the authors. We intend to perform exhaustive characterizations of BERs of the ReRAMs, with additional enrollment cycles, extended temperature cycles in the $-40\,^\circ$C to $+140\,^\circ$C range, increased buffer sizes of the number of pairs, and accelerated aging cycles. As the expected BERs will be in the $10^{-6}$ to $10^{-10}$ range, the experiments will need to run for months in order to produce statistically valid results. The cryptographic protocols and the software driving the schemes require further optimization to enhance security and mitigate various cyber-attacks. We are also aware of the need to involve independent third-party investigators to identify potential weaknesses in the proposed methods. The methods presented in this paper could be considered for the following applications:

- Per paid content delivery. A service provider can deliver several encrypted files containing information such as movies, music, apps, maps, and operating systems. The user obtains access to the files after paying a fee.
- Protected user manuals. Staged access to a prepared set of instructions for a particular task, which evolves over time, due to changes in conditions. The users receive, as needed, access codes to open a particular portion of a user manual. An example of such an application would be pilots flying a plane.
- Cooperative users. The server concurrently sends to user 2 the information needed by user 1 to retrieve a sub-key, and to user 1 the information needed by user 2 to retrieve the complementary sub-key. The full key is generated by knowledge of both sub-keys.
- Securing interconnected IoTs. Nodes of IoTs such as controlling and metering elements in a grid, home hubs, smart sensors, contain information that is stored locally and which needs to be protected constantly.
- Authentication of the server. When operating in a zero-trust environment, the server sends users information previously used to encrypt and store a session key.

**Author Contributions:** Conceptualization, and methodology, B.F.C. and S.J.; software, S.J.; validation, formal analysis, and investigation, B.F.C. and S.J.; resources, B.F.C.; data curation, B.F.C. and S.J.; writing—original draft preparation, B.F.C. and S.J.; writing—review and editing, B.F.C. and S.J.; visualization, supervision, project administration, and funding acquisition, B.F.C. All authors have read and agreed to the published version of the manuscript.

**Funding:** This research was funded by the United States Air Force Research Laboratory (AFRL) of Rome, New York, contract number 19-0437 of the Broad Agency Announcement number FA8750-19-S-7003.

**Institutional Review Board Statement:** Not applicable.

**Informed Consent Statement:** Not applicable.

**Data Availability Statement:** Not applicable.

**Acknowledgments:** The authors thank their research partners at Northern Arizona University for their support, in particular Ian Burke, Christopher Philabaum, Jack Garrard, Michael Partridge, Morgan Riggs, and Julie Heynessens. In addition, the authors thank the members of AFRL, Donald Telesca and Shelton Jacinto. Several members of Crossbar Incorporated are also recognized for their support and guidance, in particular Jo. Sung-Hyun, Hagop Nazarian, Ashish Pancholy, and Mehdi Asnaashari.

**Conflicts of Interest:** The authors declare no conflict of interest.

## Abbreviations

| | |
|---|---|
| AESAFRL | Advanced Encryption StandardUnited states Air Force Research Laboratory |
| BER | Bit Error Rate |
| CBRAM | Conductive Bridge Random Access Memory |
| CMOSCRP | Complementary Metal Oxide SiliconChallenge Response Pair |
| ECC | Error Correcting Code |
| DES | Data Encryption System |
| DRAM | Dynamic Random Access Memory |
| FRR | False Reject Rate |
| FPGA | Field Programable Gate Array |
| GPU | Graphic Processing Unit |
| HPC | High Performance Computing |
| IoTMD | Internet of ThingsMessage Digest |
| MIPS | Microprocessor without Interlocked Pipelined Stages |
| MRAM | Metal Random Access Memory |
| MUX | Multiplexer |
| NIST | National Institute of Standard and Technology |
| PKI | Public Key Infrastructure |
| PUFPW | Physical Unclonable functionPassword |
| RBC | Response Based Cryptography |
| ReRAM | Resistive Random Access Memory |
| RO | Ring Oscillator |
| RSA | Rivest Shamir Adleman code |
| SHA | Standard Hashing Algorithm |
| SRAM | Static Random Access Memory |
| XOF | Extended Output Function |
| XOR | Exclusive "OR" Gate |

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
