# Peer review of "Key Recovery for Content Protection Using Ternary PUFs Designed with Pre-Formed ReRAM"

_applsci, doi:10.3390/app12041785_

Round 1

Reviewer 1 Report

Review for: Key recovery for content protection using ternary PUFs, designed with pre-formed ReRAM – by Cambou B ., and Jain S.

Utilizing PUFs for cryptographic applications remains challenging due to the points this paper tries to tackle, such as degradation of PUF elements, which are here used to allow for key recovery of secured information. It is of interest to the research community.

Remarks:

Formatting:

Line 54, keep “section” identical. I’d prefer the big letter “S” in this case.

Line 140: MUX, write it out before abbreviating it – multiplexer (MUX),

Line 141: same goes for “RS”-Latch, write it out before abbreviating it

Line 149: […] . SRAM-based; Don’t use abbreviations at the beginning of the sentence, write it out in this case. Also check the rest of the paper on this points (e.g. Line 161 […] . PUFs ….)

Line 325 & 339: IDAccess is not written in italic anymore. Stick to one formatting style.

General formatting: 20 °C is the correct way to display the temperature, please correct it in your manuscript.

Write the H in Hamming in capital letters, it’s a name.

Open questions:

Line 89 - 94: From my understanding it should be IDaccess <- (I xor MD). This would explain the possibility to store it on the client device, and the possibility for masking of IDaccess the authors mentioned later in the manuscript. Or does the protocol require storage of plain text on the client device (e.g. sec 3.1, 3.2)? Would that open up the challenge schedule, which can be used to regenerate (in this case K’), if T & IDaccess are stored in a database on the client device? Furthermore, this would also interfere with the definition of the one-way-ness of the function: (Line 103). Or better, is one-way-ness of the function not guaranteed, if both T and IDaccess (as plain text) are known? Please elaborate

Line 228 - 229: Here the authors highlight the possibilities of RBC and how it can benefit from powerful hardware. However, this explanation somehow is counterintuitive for session key recovery, as the load is now required on the client device to recover K’’. Here either be clear in describing how it is beneficial, how hardware scaling helps in this specific case (on server side), or even leave it as it is counterintuitive that you want to use such a thing on the constrained IoT device, which will most likely hold low-cost peripherals.

Regarding that point, how much power was required for the reported set of thirty 256-bit long key cycles with SRAM and ReRAM using your strategy on the demo boards? I also recall it was the light implementation. What would be the expected power consumption using the full, secure protocol which relies on communication with the server. I think this is interesting information as it increases comparability and improves general distinguishability and selection of appropriate protocol for the target application.

Line 448 (Subsect. 5.2): As the expected minimum entropy of the PUF response R, here K, depends on internal biases of e.g. the SRAM cell arrays or the ReRAM arrays and the readout-electronics I’d like to know if there are already works regarding PUF metrics – especially on the ReRAM arrays conducted. Such as uniqueness, reliability, bit aliasing to name but a few.

Line 455: What does “injected currents between 100 nA and 800 nA” mean? Is this a fixed, say ramping scheme or a fixed current stepsize, which is part of the challenge OR something that is variable and selects the matching currents for each CRP. For simplicity the challenge schedule should best be a fixed routine, as else the challenge per PUF needs to change, which introduces other factors to be considered. Please elaborate.

Author Response

See file enclosed...thank you

Reviewer 2 Report

Many acronyms are used in this study, it is better to list all of them in a single table.

Line 79: "Such a function has the following properties" better to number these properties or may use subheading style

better to use standard mathematical symbols 

font size (text inside some boxes) of figure 1-6 is large, need to be adjusted according to the size of figures

Add few lines between heading 6 and its sub heading 1

Line number 673-676 are subpoints of bullet 2? why their style is different

Future work may be added as a separate section  for providing clear directions to researchers

Author Response

See file enclosed...thank you
